# AUTOREGRESSIVE LATENT VIDEO PREDICTION WITH HIGH-FIDELITY IMAGE GENERATOR

## ABSTRACT

Video prediction is an important yet challenging problem; burdened with the tasks of generating future frames and learning environment dynamics. Recently, autoregressive latent video models have proved to be a powerful video prediction tool, by separating the video prediction into two sub-problems: pre-training an image generator model, followed by learning an autoregressive prediction model in the latent space of the image generator. However, successfully generating high-fidelity and high-resolution videos has yet to be seen. In this work, we investigate how to train an autoregressive latent video prediction model capable of predicting high-fidelity future frames with minimal modification to existing models, and produce high-resolution (256x256) videos. Specifically, we scale up prior models by employing a high-fidelity image generator (VQ-GAN) with a causal transformer model, and introduce additional techniques of top-$k$ sampling and data augmentation to further improve video prediction quality. Despite the simplicity, the proposed method achieves competitive performance to state-of-the-art approaches on standard video prediction benchmarks with fewer parameters, and enables high-resolution video prediction on complex and large-scale datasets. Videos are available at the anonymized website https://sites.google.com/view/harp-anonymous.

## 1 INTRODUCTION

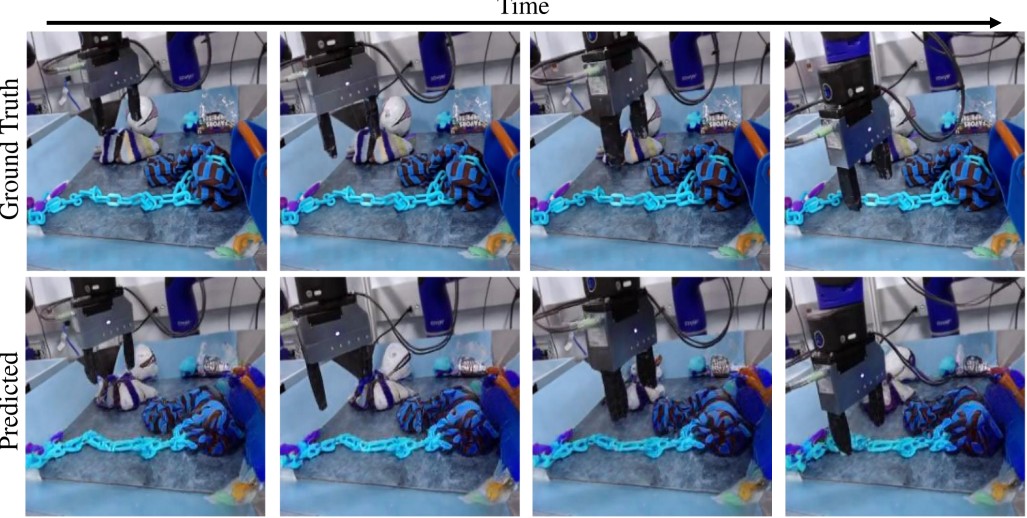

Figure 1: Selcted $256 \times 256$ video sample generated by HARP on RoboNet (Dasari et al., 2019).

Video prediction can enable agents to learn useful representations for predicting the future consequences of the decisions they make, which is crucial for solving the tasks that require long-term planning, including robotic manipulation (Finn & Levine, 2017; Kalashnikov et al., 2018) and autonomous driving (Levinson et al., 2011; Xu et al., 2017). Despite the recent advances in improving the quality of video prediction (Finn et al., 2016; Lotter et al., 2017; Liang et al., 2017; Babaeizadeh et al., 2018; Denton & Fergus, 2018; Lee et al., 2018; Byeon et al., 2018; Kumar et al., 2020; Weissenborn et al., 2020; Babaeizadeh et al., 2021), learning an accurate video prediction model remains

notoriously difficult problem and requires a lot of computing resources, especially when the inputs are video sequences with high-resolution (Castrejon et al., 2019; Villegas et al., 2019; Clark et al., 2019; Luc et al., 2020; Walker et al., 2021). This is because the video prediction model should excel at both tasks of generating high-fidelity images and learning the dynamics of environments, though each task itself is already a very challenging problem.

Recently, autoregressive latent video prediction methods (Rakhimov et al., 2021; Yan et al., 2021) have been proposed to improve the efficiency of video prediction, by separating video prediction into two sub-problems: first pre-training an image generator (*e.g.,* VQ-VAE; Oord et al. 2017), and then learning the autoregressive prediction model (Weissenborn et al., 2020; Chen et al., 2020) in the latent space of the pre-trained image generator. However, the prior works are limited in that they only consider relatively low-resolution videos (up to $128 \times 128$ pixels) for demonstrating the efficiency of the approach; it is questionable that such experiments can fully demonstrate the benefit of operating in the latent space of image generator instead of high-dimensional pixel-channel space.

In this paper, we present **H**igh-fidelity **A**uto**R**egressive latent video **P**rediction (HARP), which scales up the previous autoregressive latent video prediction methods for high-fidelity video prediction. The main principle for the design of HARP is simplicity: we improve the video prediction quality with minimal modification to existing methods. First, for image generation, we employ a high-fidelity image generator, *i.e.,* vector-quantized generative adversarial network (VQ-GAN; Esser et al. 2021). This improves video prediction by enabling high-fidelity image generation (up to $256 \times 256$ pixels) on various video datasets. Then a causal transformer model (Chen et al., 2020), which operates on top of discrete latent codes, is trained to predict the discrete codes from VQ-GAN, and autoregressive predictions made by the transformer model are decoded into future frames at inference time. Moreover, motivated by the sampling techniques widely-used in language generation for making coherent and diverse predictions, we propose to utilize top-$k$ sampling (Fan et al., 2018) that draws the next discrete code from the $k$-most probable codes. Since the number of discrete codes the autoregressive model has to predict is very large, *e.g.,* 6,400 codes on KITTI driving dataset (Geiger et al., 2013), we find that discarding rare discrete codes helps the model predict diverse but high-quality videos, without any change to the training procedure.

We highlight the main contributions of this paper below:

- We show that our autogressive latent video prediction model, HARP, can predict high-resolution ($256 \times 256$ pixels) future frames on simulated robotics dataset (*i.e.,* Meta-World; Yu et al. 2020) and large-scale real-world robotics dataset (*i.e.,* RoboNet; Dasari et al. 2019).

- We show that HARP can leverage the image generator pre-trained on ImageNet (Deng et al., 2009) for training a high-resolution video prediction model on complex, large-scale Kinetics-600 dataset (Carreira et al., 2018), significantly reducing the training cost.

- HARP achieves competitive or superior performance to prior state-of-the-art video prediction models with large end-to-end networks on widely-used BAIR Robot Pushing (Ebert et al., 2017) and KITTI driving (Geiger et al., 2013) video prediction benchmarks.

- We also show that the pre-trained representations of HARP can be useful for learning multi-task imitation learning agent on Meta-World MT50 benchmark (Yu et al., 2020).

## 2 RELATED WORK

**Video prediction.** Video prediction aims to predict the future frames conditioned on images (Michalski et al., 2014; Ranzato et al., 2014; Srivastava et al., 2015; Vondrick et al., 2016; Lotter et al., 2017), texts (Wu et al., 2021b), and actions (Oh et al., 2015; Finn et al., 2016), which would be useful for several applications, *e.g.,* model-based RL (Hafner et al., 2019; Kaiser et al., 2020; Rybkin et al., 2021), and simulator development (Kim et al., 2020; 2021). Various video prediction models have been proposed with different approaches, including generative adversarial networks (GANs; Goodfellow et al. 2014) known to generate high-fidelity images by introducing adversarial discriminators that also considers temporal or motion information (Aigner & Körner, 2018; Jang et al., 2018; Kwon & Park, 2019; Clark et al., 2019; Luc et al., 2020), latent video prediction models that operates on the latent space (Babaeizadeh et al., 2018; Denton & Fergus, 2018; Lee et al., 2018; Villegas et al., 2019; Wu et al., 2021a; Babaeizadeh et al., 2021), and autoregressive video

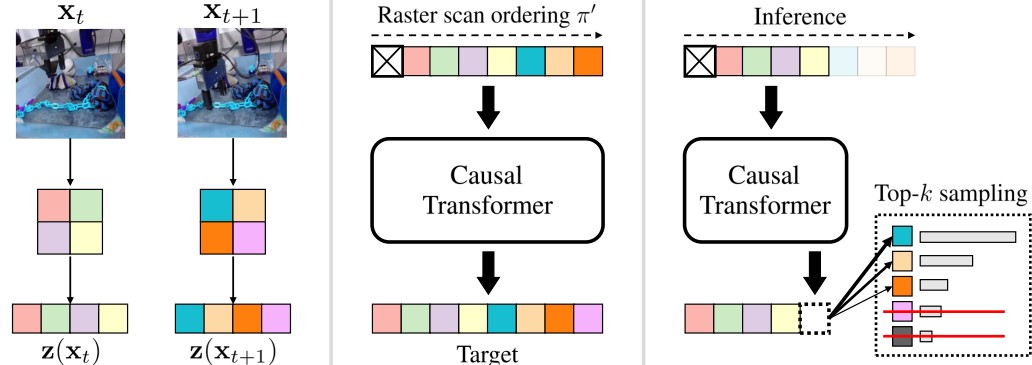

Figure 2: Illustration of our approach. We first train a VQ-GAN model that encodes frames into discrete latent codes. Then the discrete codes are flattened following the raster scan order, and a causal transformer model is trained to predict the next discrete codes in an autoregressive manner. At inference time, we use top-$k$ sampling that only draws the next discrete code from top-$k$ probable codes, in order to predict diverse but high-quality videos.

prediction models that operates on pixel space by predicting the next pixels in an autoregressive way (Kalchbrenner et al., 2017; Reed et al., 2017; Weissenborn et al., 2020).

**Autoregressive latent video prediction.**   Most closely related to our work are autoregressive latent video prediction models that separate the video prediction problem into image generation and dynamics learning. Walker et al. (2021) proposed to learn a hierarchical VQ-VAE (Razavi et al., 2019) that extracts multi-scale hierarchical latents then train SNAIL blocks (Chen et al., 2018) that predict hierarchical latent codes, enabling high-fidelity video prediction. However, this involves a complicated training pipeline and a video-specific architecture, which limits its applicability. As simple alternatives, Rakhimov et al. (2021); Yan et al. (2021) proposed to first learn a VQ-VAE (Oord et al., 2017) and train a causal transformer with 3D self-attention (Weissenborn et al., 2020) and factorized 2D self-attention (Child et al., 2019), respectively. These approaches, however, are limited in that they only consider low-resolution videos. We instead present a simple high-resolution video prediction method that incorporates the strengths of both prior approaches.

## 3   PRELIMINARIES

We consider the standard video prediction framework where the goal is to predict the future frames conditioned on the initial frames of a video. Specifically, conditioned on the first $c$ frames of a video $\mathbf{x}_{<c} = (\mathbf{x}_0, \mathbf{x}_1, ..., \mathbf{x}_{c-1})$, we aim to learn a video prediction model that predicts the future frames $\mathbf{x}_{c:T} = (\mathbf{x}_c, ..., \mathbf{x}_{T-1})$, where $\mathbf{x}_t \in \mathbb{R}^{H \times W \times N_{ch}}$ is the frame at timestep $t$. Optionally, one can also consider conditioning the prediction model on actions $\mathbf{a} = (\mathbf{a}_0, ..., \mathbf{a}_{T-1})$ that the agents in the video would take, *i.e.,* action-conditioned video prediction.

**Autoregressive video prediction model.**   Motivated by the recent success of pixel-level autoregressive models on image generation (Menick & Kalchbrenner, 2018), Weissenborn et al. (2020) introduced an autoregressive video prediction model that approximates the distribution of a video in a pixel-channel space. Specifically, given a video $\mathbf{x} \in \mathbb{R}^{T \times H \times W \times N_{ch}}$, the joint distribution over pixels conditioned on the first $c$ frames is modelled as the product of channel intensities $N_{ch}$ and all $N_p = T \cdot H \cdot W$ pixels except $N_c = c \cdot H \cdot W$ pixels of conditioning frames:

$$p(\mathbf{x}_{c:T} \,|\, \mathbf{x}_{<c}) = \prod_{i=N_c-1}^{N_p-1} \prod_{k=0}^{N_{ch}-1} p(\mathbf{x}_{\pi(i)}^k | \mathbf{x}_{\pi(<i)}, \mathbf{x}_{\pi(i)}^{<k}), \qquad (1)$$

where $\pi$ is a raster-scan ordering over all pixels from the video (we refer to Weissenborn et al. (2020) for more details on the case where $\pi$ is the combination of a subscale and raster-scan ordering since we only utilize raster-scan ordering for our approach), $\mathbf{x}_{\pi(<i)}$ is all pixels before $\mathbf{x}_{\pi(i)}$, $\mathbf{x}_{\pi(i)}^k$ is the $k$-th channel intensity of the pixel $\mathbf{x}_{\pi(i)}$, and $\mathbf{x}_{\pi(i)}^{<k}$ is all channel intensities before $\mathbf{x}_{\pi(i)}^k$.

**Vector quantized variational autoencoder.** VQ-VAE (Oord et al., 2017) consists of an encoder that compresses images into discrete representations, and a decoder that reconstructs images from these discrete representations. Both encoder and decoder share a codebook of prototype vectors which are also learned throughout training. Formally, given an image $x \in \mathbb{R}^{H \times W \times N_{ch}}$, the encoder $E$ encodes $x$ into a feature map $z_e(x) \in \mathbb{R}^{H' \times W' \times N_z}$ that consists of a series of latent vectors $z_{\pi'(i)}(x) \in \mathbb{R}^{N_z}$, where $\pi'$ is a raster-scan ordering of the feature map $z_e(x)$ of size $|\pi'| = H' \cdot W'$. Then $z_e(x)$ is quantized to discrete representations $z_q(x) \in \mathbb{R}^{|\pi'| \times N_z}$ based on the distance of latent vectors $z_{\pi'(i)}(x)$ to the prototype vectors in the codebook $C = \{e_k\}_{k=1}^{K}$ as follows:

$$z_q(x) = (e_{q(x,1)}, e_{q(x,2)}, \cdots, e_{q(x,|\pi'|)}), \quad \text{where } q(x, i) = \operatorname*{arg\,min}_{k \in [K]} \|z_{\pi'(i)}(x) - e_k\|_2, \quad (2)$$

where $[K]$ is the set $\{1, \cdots, K\}$. Then the decoder $G$ learns to reconstruct $x$ from discrete representations $z_q(x)$. The VQ-VAE is trained by minimizing the following objective:

$$\mathcal{L}_{\texttt{VQVAE}}(x) = \underbrace{\|x - G(z_q(x))\|_2^2}_{\mathcal{L}_{\texttt{recon}}} + \underbrace{\|sg\,[z_e(x)] - z_q(x)\|_2^2}_{\mathcal{L}_{\texttt{codebook}}} + \underbrace{\beta \cdot \|sg\,[z_q(x)] - z_e(x)\|_2^2}_{\mathcal{L}_{\texttt{commit}}}, \quad (3)$$

where the operator $sg$ refers to a stop-gradient operator, $\mathcal{L}_{\texttt{recon}}$ is a reconstruction loss for learning representations useful for reconstructing images, $\mathcal{L}_{\texttt{codebook}}$ is a codebook loss to bring codebook representations closer to corresponding encoder outputs $h$, and $\mathcal{L}_{\texttt{commit}}$ is a commitment loss weighted by $\beta$ to prevent encoder outputs from fluctuating frequently between different representations.

**Vector quantized generative adversarial network.** VQ-GAN (Esser et al., 2021) is a variant of VQ-VAE that (a) replaces the $\mathcal{L}_{\texttt{recon}}$ in (3) by a perceptual loss $\mathcal{L}_{\texttt{LPIPS}}$ (Zhang et al., 2018), and (b) introduces an adversarial training scheme where a patch-level discriminator $D$ (Isola et al., 2017) is trained to discriminate real and generated images by maximizing following loss:

$$\mathcal{L}_{\texttt{GAN}}(x) = [\log D(x) + \log(1 - D(G(z_q(x))))]. \quad (4)$$

Then, the objective for training the VQ-GAN model is defined as:

$$\min_{E,G,C} \max_{D} \mathbb{E}_{x \sim p(x)} \left[ (\mathcal{L}_{\texttt{LPIPS}} + \mathcal{L}_{\texttt{codebook}} + \mathcal{L}_{\texttt{commit}}) + \lambda \cdot \mathcal{L}_{\texttt{GAN}} \right], \quad (5)$$

where $\lambda = \frac{\nabla_{G_L}[\mathcal{L}_{\texttt{LPIPS}}]}{\nabla_{G_L}[\mathcal{L}_{\texttt{GAN}}] + \delta}$ is an adaptive weight, $\nabla_{G_L}$ is the gradient of the inputs to the last layer of the decoder $G_L$, and $\delta = 10^{-6}$ is a scalar introduced for numerical stability.

# 4 METHOD

We present HARP, a video prediction model capable of predicting high-fidelity future frames. Our method is designed to fully exploit the benefit of autoregressive latent video prediction model that separates the video prediction into image generation and dynamics learning. Specifically, we consider the combination of (a) the recently introduced high-fidelity image generator (Esser et al., 2021) and (b) an autoregressive latent video prediction model (Oord et al., 2017; Rakhimov et al., 2021; Walker et al., 2021; Yan et al., 2021) that operates on top of the pre-trained image generator. The full architecture of HARP is illustrated in Figure 2.

## 4.1 HIGH-FIDELITY IMAGE GENERATOR

We consider the VQ-GAN model (Esser et al., 2021) that has proven to be effective for high-resolution image generation as our image generator (see Section 3 for the formulation of VQ-GAN). Similar to the motivation of Tian et al. (2021) that utilizes a pre-trained image generator in the context of video synthesis, we first pre-train the image generator then freeze the model throughout training to improve the efficiency of learning video prediction models. The notable difference to a prior work that utilize 3D convolutions to temporally downsample the video for efficiency (Yan et al., 2021) is that our image generator operates on single images; hence our image generator solely focus on improving the quality of generated images. Importantly, this enables us to utilize the VQ-GAN model pre-trained on a wide range of natural images, *e.g.,* ImageNet, without training the image generator on the target datasets, which can significantly reduce the training cost of high-resolution video prediction model, Also, the representations from our video prediction model can be easily transferred to downstream tasks that require fine-grained control at each timestep, *e.g.,* imitation learning (see Section 5.3 for supporting experimental results on multi-task imitation learning).

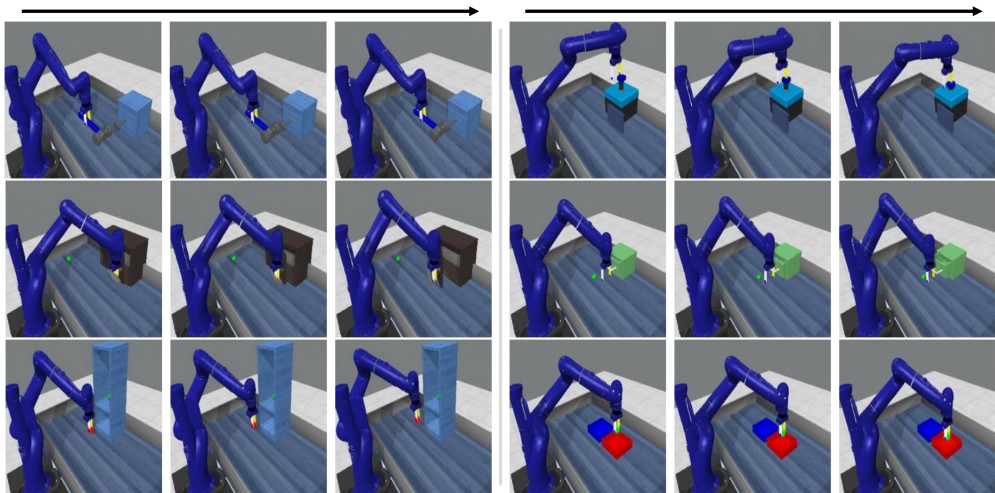

Figure 3: $256 \times 256$ future frames predicted by HARP on action-free Meta-World (Yu et al., 2020). Using videos from 50 different asks, we train a single model to predict the 11 frames conditioned on the first frame without leveraging task-specific information (*e.g.,* task index). We observe that the model can accurately predict the high-resolution future frames of diverse tasks. More videos predicted by HARP are available in Appendix D.

## 4.2    AUTOREGRESSIVE LATENT VIDEO PREDICTION MODEL

To leverage the VQ-GAN model for video prediction, we utilize the autoregressive latent video prediction architecture that operates on top of the discrete codes extracted from a video $\mathbf{x}$. Specifically, we extract the discrete codes $\mathbf{z}(\mathbf{x}) = (\mathbf{z}(\mathbf{x}_1), ..., \mathbf{z}(\mathbf{x}_T))$ using the pre-trained VQ-GAN, where $\mathbf{z}(\mathbf{x}_t) = (q_{(\mathbf{x}_t,1)}, q_{(\mathbf{x}_t,2)}, ..., q_{(\mathbf{x}_t,|\pi'|)})$ is the discrete code extracted from the frame $\mathbf{x}_t$ as in (2). Then, instead of modelling the distribution of video $p(\mathbf{x})$ in the pixel-channel space as in (1), we learn the distribution of the video in the discrete latent representation space:

$$p(\mathbf{z}(\mathbf{x}_{c:T}|\mathbf{x}_{<c})) = \prod_{i=0}^{N_d-1} p(\mathbf{z}_{\pi'(i)}(\mathbf{x})|\mathbf{z}_{\pi'(<i)}(\mathbf{x})), \tag{6}$$

where $N_d = (T - C) \cdot H' \cdot W'$ is the total number of codes from $\mathbf{x}_{c:T}$. While the specific implementation for modelling $p(\mathbf{z}(\mathbf{x}))$ differs in prior works (Oord et al., 2017; Rakhimov et al., 2021; Walker et al., 2021; Yan et al., 2021), due to its simplicity, we utilize the causal transformer architecture (Yan et al., 2021) where the output logits from input codes are trained to predict the next discrete codes. We remark that our approach is also compatible with other architectures.

## 4.3    ADDITIONAL TECHNIQUES

**Top-k sampling.**    To improve the video prediction quality of latent autoregressive models whose outputs are sampled from the probability distribution over a large number of discrete codes, we utilize the top-$k$ sampling (Fan et al., 2018) that randomly samples the output from the top-$k$ probable discrete codes. By preventing the model from sampling rare discrete codes from the long-tail of a probability distribution and predicting future frames conditioned on such discrete codes, we find that top-$k$ sampling improves video prediction quality, especially given that the number of discrete encodings required for future prediction is very large, *e.g.,* 2,560 on RoboNet (Dasari et al., 2019) up to 6,400 on KITTI dataset (Geiger et al., 2013) in our experimental setup.

**Data augmentation.**    We also investigate how data augmentation can be useful for improving the performance of autoregressive latent video prediction models. Since the image generator model is not trained with augmentation, we utilize a weak augmentation to avoid the instability coming from aggressive transformation of input frames, *i.e.,* translation augmentation that moves the input images by $m$ pixels along the X or Y direction.

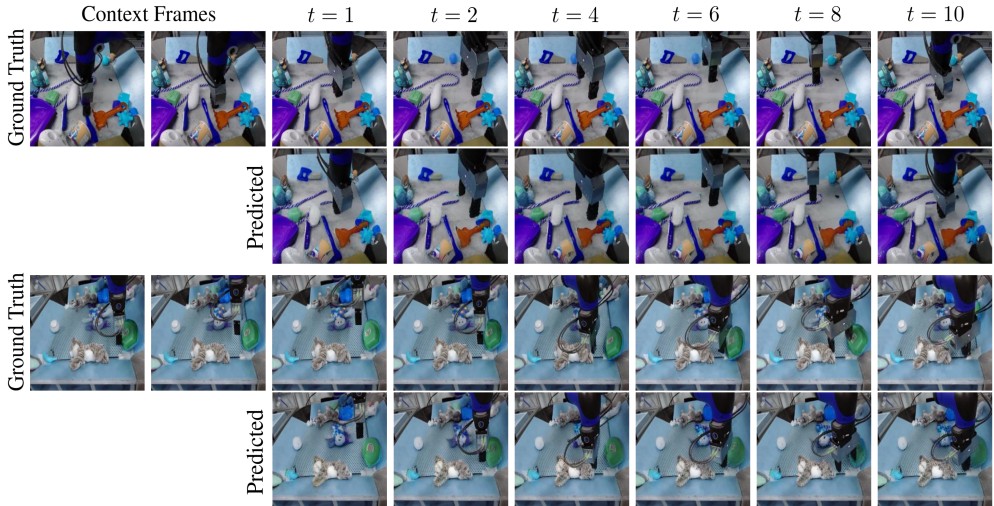

Figure 4: $256 \times 256$ future frames predicted by HARP on action-conditioned RoboNet (Dasari et al., 2019). The model predicts the next ten frames conditioned on the first two frames and the future ten actions. We observe that our model can predict the movement of a robot arm surrounded by various objects of different colors and shapes. More videos predicted by HARP are available in Appendix E.

## 5 EXPERIMENTS

We design our experiments to investigate the following:

- Can HARP predict high-resolution future frames (up to $256 \times 256$ pixels) on various video datasets with different characteristics?
- How does HARP compare to state-of-the-art methods with large end-to-end networks on standard video prediction benchmarks in terms of quantitative evaluation?
- How does the proposed techniques affect the performance of HARP?
- Can HARP be transferred to solve multi-task imitation learning tasks?

### 5.1 HIGH-RESOLUTION VIDEO PREDICTION

**Implementation.** We utilize up to 8 Nvidia 2080Ti GPU and 20 CPU cores for training each model. For training VQ-GAN (Esser et al., 2021), we first train the model without a discriminator loss $\mathcal{L}_{\texttt{GAN}}$, and then continue the training with the loss following the suggestion of the authors. For all experiments, VQ-GAN downsamples each frame into $16 \times 16$ latent codes, *i.e.,* by a factor of 4 for frames of size $64 \times 64$ frames, and 16 for frames of size $256 \times 256$. For training a transformer model, the VQ-GAN model is frozen so that its parameters are not updated. We use Sparse Transformers (Child et al., 2019) as our transformer architecture to accelerate the training. As for hyperparameter, we use $k = 10$ for sampling at inference time, but no data augmentation for high-resolution video prediction experiments. We report more detailed implementation details in Appendix A.

**Meta-World experiments.** To demonstrate that our method can predict high-resolution videos ($256 \times 256$ pixels), we first use action-free Meta-World dataset (Yu et al., 2020) consisting of 2,500 demonstrations from 50 different robotics manipulation tasks, which we collected using the deterministic scripted policies[1]. Specifically, we train a single model to predict future videos of all 50 tasks, without leveraging task-specific information, such as task index. For evaluation, we use 10% of the demonstrations as a held-out test dataset. As shown in Figure 3, our model can accurately predict the high-resolution future frames of diverse tasks, capturing all the small details. This shows that our model can effectively learn all the information of multiple tasks required for predicting future frames. We also remark that such representations can be useful to improve the performance of imitation learner (see Section 5.3 for supporting experimental results).

**RoboNet experiments.** Now we investigate how our model works on large-scale, real-world RoboNet dataset (Dasari et al., 2019) consisting of more than 15 million frames. While prior works successfully trained a video prediction model with $64 \times 64$ videos (Wu et al., 2021a; Babaeizadeh

---

[1]The dataset is available at: https://shorturl.at/acnxM

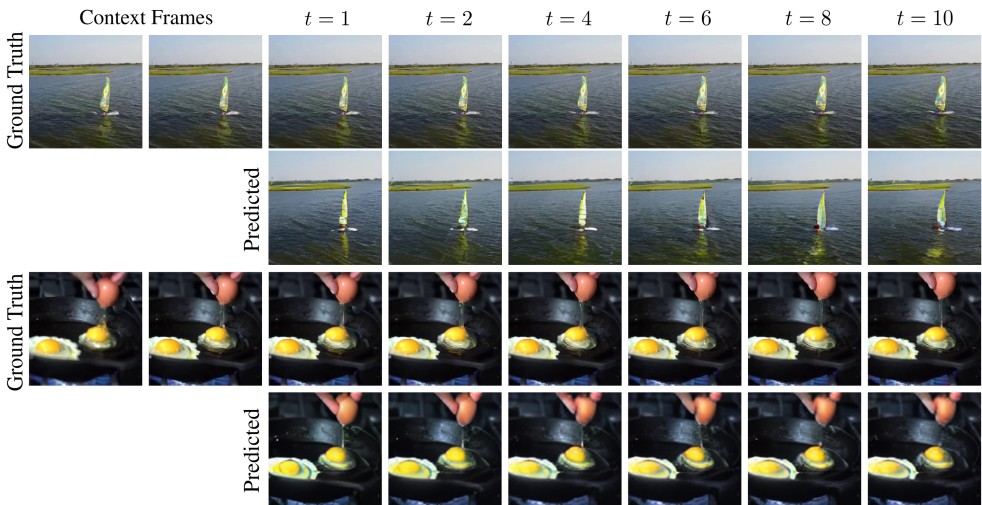

Figure 5: $256 \times 256$ future frames predicted by HARP trained on Kinteics-600 dataset (Carreira et al., 2018) using ImageNet pre-trained VQ-GAN model. Videos are licensed under CC-BY and attribution can be found in Appendix G. More videos predicted by HARP are available in Appendix F.

et al., 2021), we show that our model can predict high-resolution $256 \times 256$ videos even with fewer number of parameters than the one used in the prior works for predicting $64 \times 64$ videos. Specifically, we first train a VQ-GAN model with 91.5M parameters, and then train a 12-layer causal transformer model with 74.2M parameters that predicts future 10 frames conditioned on first two frames and future ten actions. Total number of parameters is 165.7M, which is smaller than 303.3M of FitVid (Babaeizadeh et al., 2021) that predicts $64 \times 64$ videos. Figure 1 and Figure 4 show the predicted frames on the held-out test video, where the model predicts the high-resolution future frames where a robot arm is moving around various objects of different colors and shapes.

**Kinetics-600 experiments using ImageNet pre-trained VQ-GAN.** Finally, we consider a very complex, large-scale Kinetics-600 dataset (Carreira et al., 2018) consisting of more than 400,000 videos, which requires a large amount of computing resources for training even on $64 \times 64$ resolution (Clark et al., 2019; Luc et al., 2020). To avoid the prohibitively expensive training cost of high-resolution video prediction models on this dataset and fully exploit the benefit of employing a high-fidelity image generator, we utilize the VQ-GAN model pre-trained on ImageNet dataset (Deng et al., 2009). [2] As we only train the transformer model for video prediction, this enables us to train a high-resolution video prediction model in a very efficient manner. Specifically, we train the transformer model for 60,000 steps on training dataset, which takes less than a day using our machine. As shown in Figure 5, our model can predict future frames on the test videos[3], which demonstrates that leveraging the large image generator pre-trained on a wide range of natural images can be a promising recipe for efficient video prediction on high-resolution, large-scale video datasets.

## 5.2 COMPARATIVE EVALUATION ON STANDARD BENCHMARKS

**Datasets.** For quantitative evaluation, we first consider the BAIR robot pushing dataset (Ebert et al., 2017) consisting of roughly 40k training and 256 test videos. We consider action-free setup, hence video prediction models should be stochastic for predicting the diverse possible movement of a robot arm and objects. Following the setup in prior works (Clark et al., 2019; Weissenborn et al., 2020; Luc et al., 2020; Yan et al., 2021), we predict 15 future frames conditioned on one frame. We also evaluate our method on KITTI driving dataset (Geiger et al., 2013), where the training and test datasets are split by following the setup in Lotter et al. (2017). As the KITTI dataset is relatively small-scale compared to other datasets, *i.e.,* 57 training videos, it provides a good testbed for investigating the effect of data augmentation. For hyperparameters, we use $k = 10$ for both datasets and data augmentation with $m = 4$ is only applied to KITTI as there was no sign of overfitting on BAIR dataset. For a fair comparison, we follow the setup of Villegas et al. (2019), where (i) a model is trained to predict future ten frames conditioned on five frames and evaluated to predict future 25 frames conditioned on five frames, and (ii) test dataset consists of 148 video clips constructed by extracting 30-frame clips and skipping every 5 frames.

---

[2]https://github.com/CompVis/taming-transformers

[3]We collected videos covered by CC-BY license, which are available to put the frames on a paper.

Table 1: Quantitative evaluation on (a) BAIR Robot Pushing (Ebert et al., 2017) and (b) KITTI driving dataset (Geiger et al., 2013). We observe that HARP can achieve competitive performance to state-of-the-art methods with large end-to-end networks on these benchmarks.

(a) BAIR Robot Pushing

| Method[4] | Params | FVD ($\downarrow$) |
|---|---|---|
| LVT | 50M | 125.8 |
| SAVP | 53M | 116.4 |
| DVD-GAN-FP | —[†] | 109.8 |
| VideoGPT | 82M | 103.3 |
| TrIVD-GAN-FP | —[†] | 103.3 |
| Video Transformer | 373M | 94.0 |
| FitVid | 302M | **93.6** |
| HARP (ours) | 89M | 99.3 |

(b) KITTI

| Method[4] | Params | FVD ($\downarrow$) | LPIPS ($\downarrow$) |
|---|---|---|---|
| SVG | 298M | 1217.3 | 0.327 |
| GHVAE | 599M | 552.9 | 0.286 |
| FitVid | 302M | 884.5 | 0.217 |
| HARP (ours) | 89M | **482.9** | **0.191** |

[†] Not available

Table 2: FVD scores of HARP with varying (a) the number of codes to use for top-$k$ sampling, (b) number of layers, and (c) magnitude $m$ of data augmentation.

(a) Effects of $k$

| Dataset | $k$ | FVD ($\downarrow$) |
|---|---|---|
| BAIR | No top-$k$ | 104.4 |
| | 100 | 103.6 |
| | 10 | **99.3** |
| KITTI | No top-$k$ | 578.1 |
| | 100 | 557.7 |
| | 10 | **482.9** |

(b) Effects of layers

| Dataset | Layers | FVD ($\downarrow$) |
|---|---|---|
| BAIR | 6 | 111.8 |
| | 12 | **99.3** |
| KITTI | 6 | 520.1 |
| | 12 | **482.9** |

(c) Effects of $m$

| Dataset | $m$ | FVD ($\downarrow$) |
|---|---|---|
| KITTI | 0 | 980.1 |
| | 2 | 497.0 |
| | 4 | **482.9** |
| | 8 | 523.4 |

**Metrics.** We use two evaluation metrics: Learned Perceptual Image Patch Similarity (LPIPS; Zhang et al. 2018), a frame-wise metric designed to better represent the human perceptual similarity of two frames compared to traditional metrics (Wang et al., 2004; Huynh-Thu & Ghanbari, 2008), and Frèchet Video Distance (FVD; Unterthiner et al. 2018), a dynamics-based evaluation metric known to be better correlated with the human evaluation compared to frame-wise evaluation metrics. FVD is computed by comparing the summary statistics of I3D network trained on Kinetics-400 dataset (Carreira & Zisserman, 2017), and LPIPS is computed using the features from AlexNet (Krizhevsky et al., 2012). For comparison with the scores reported in prior works, we exactly follow the evaluation setup in Villegas et al. (2019) and Babaeizadeh et al. (2021) that samples 100 future videos for each ground-truth test video, then reports the best score over 100 videos for LPIPS, and the score using all videos for FVD, with the batch size of 256 for BAIR and 148 for KITTI.

**Results.** Table 1 shows the performances of our method and baselines on test sets of BAIR Robot Pushing and KITTI driving dataset. We observe that our model achieves competitive or superior performance to state-of-the-art methods with large end-to-end networks, *e.g.,* HARP outperforms FitVid with 302M parameters on KITTI driving dataset. Our model successfully extrapolates to unseen number of future frames (*i.e.,* 25) instead of 10 future frames used in training on KITTI dataset. This implies that transformer-based video prediction models can also predict arbitrary number of frames at inference time. In the case of BAIR dataset, HARP achieves the similar performance of FitVid with 302M parameters, even though our method only requires 89M parameters. We provide videos predicted by HARP on BAIR and KITTI datasets in Appendix C.

**Analysis.** We investigate how the top-$k$ sampling, number of layers, and magnitude $m$ of data augmentation affect the performance. Table 2a shows that smaller $k$ leads to better performance, implying that the proposed top-$k$ sampling is effective for improving the performance by discarding rare discrete codes that might degrade the prediction quality at inference time. As shown in Table 2b, we observe that more layers leads to better performance on BAIR dataset, which implies our model can be further improved by scaling up the networks. Finally, we find that (i) data augmentation on KITTI dataset is important for achieving strong performance, similar to the observation of Babaeizadeh et al. (2021), and (ii) too aggressive augmentation leads to worse performance. We provide the learning curves with and without augmentation in Appendix B.

---

[4]Baselines are SVG (Villegas et al., 2019), GHVAE (Wu et al., 2021a), FitVid (Babaeizadeh et al., 2021), LVT (Rakhimov et al., 2021), SAVP (Lee et al., 2018), DVD-GAN-FP (Clark et al., 2019), VideoGPT (Yan et al., 2021), TrIVD-GAN-FP (Luc et al., 2020), and Video Transformer (Weissenborn et al., 2020).

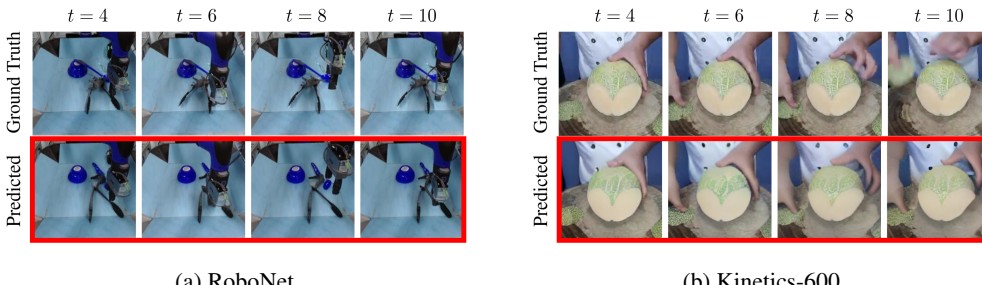

(a) RoboNet  (b) Kinetics-600

Figure 6: Failure cases in our experiments. (a) Interaction with the objects is ignored. (b) The model repeats the first frame while a person is moving right in the ground-truth frames.

### 5.3 Fine-tuning HARP for multi-task imitation learning

**Setup.** In order to demonstrate that the pre-trained representations from HARP can be useful for solving downstream tasks, we evaluate the imitation learning performance of fine-tuned HARP on MT-50 benchmark from Meta-World. Specifically, we take the pre-trained HARP model (see Figure 3 for the video predictions from this model), and fine-tune the model to predict expert actions by introducing a policy network on top of the transformer model. For comparative evaluation, we consider three baselines: (a) VQ-Transformer, which shares the same architecture with HARP but trained from scratch, (b) CNN-LSTM, which extracts features using convolutional neural networks (CNN) and LSTM networks, and (c) CNN-Transformer, that utilizes transformer networks instead of LSTM networks. For training and evaluation, we use the same training and test dataset used for video prediction experiments. We report the average success rate over 10 trials for each task. More details are available in Appendix A.

**Results.** Table 3 shows the performance of imitation learning policies on MT50 test environments. We first observe that VQ-Transformer, which has a same architecture with HARP but trained from scratch, completely fails to solve the tasks. This shows the difficulty of training useful representations with fixed discrete codes as inputs. However, fine-tuned HARP model successfully outperforms other baselines because pre-trained representations contain useful information for long-term reasoning. This demonstrates that video prediction with HARP can be an effective self-supervised learning scheme for solving various control tasks.

Table 3: Average success rate on MT-50 test environments. The results report the mean and standard deviation over five runs.

| Method | Success rate |
|---|---|
| VQ-Transformer | $0.0\pm_{0.0}$ |
| CNN-LSTM | $0.41\pm_{0.057}$ |
| CNN-Transformer | $0.50\pm_{0.055}$ |
| HARP (ours) | $\mathbf{0.58}\pm_{\mathbf{0.038}}$ |

## 6 Discussion

In this work, we present HARP, a high-fidelity autoregressive latent video prediction model. By employing a high-fidelity image generator and utilizing top-$k$ sampling at inference time, HARP can predict high-resolution future frames, and achieve competitive performance to state-of-the-art video prediction methods with large end-to-end networks. We also show that HARP can leverage the image generator pre-trained on a wide range of natural images for video prediction, similar to the approach in the context of video synthesis (Tian et al., 2021). We hope this work inspires more investigation into leveraging a pre-trained image generator for video prediction, which can significantly reduce the cost for training a high-resolution video prediction model by building on the recent success of high-fidelity image generation (Oord et al., 2017; Razavi et al., 2019; Esser et al., 2021; Child, 2020; Ho et al., 2020; Karras et al., 2020; Dhariwal & Nichol, 2021).

Finally, we report the failure cases of video prediction with HARP and discuss the possible extensions to resolve the issue. A common failure case for video prediction on RoboNet dataset is ignoring the interaction between a robot arm and objects. For example, in Figure 6a, our model ignores the objects and only predicts the movement of a robot arm. On the other hand, common failure case for Kinetics-600 is a degenerate video prediction, where a model just repeats the conditioning frame without predicting the future, as shown in Figure 6b. These failure cases might be resolved by training more larger networks similar to the observation in the field of natural language processing, *e.g.,* GPT-3 (Brown et al., 2020), or might necessitate a new architecture for addressing the complexity of training autoregressive latent prediction models on video datasets.

## ETHICS STATEMENT

While video prediction can be useful for various applications including robotic manipulation and autonomous driving, it might be misused by malicious users for unethical purposes, *e.g.,* fake videos accusing politicians or sexual videos of any individuals. As our work introduces a method for generating more high-resolution future frames, our method may improve the chance of such videos being recognized as real videos. For this reason, in addition to developing a video prediction method that generates more realistic frames, it is important to be aware of potential problems and develop a method to detect generated videos (Gentine et al., 2018).

## REPRODUCIBILITY STATEMENT

We describe the implementation and evaluation details in Section 5 and Appendix A. We also provide our code in the supplementary material.

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

# A EXPERIMENTAL SETUP

## A.1 DATASETS

**Meta-World.**    Meta-World (Yu et al., 2020) is a robotics manipulation simulator that supports 50 different tasks. For our experiments, we collect 50 demonstrations for each task using the deterministic scripted policies.[5], hence we use total 2500 demonstrations. For evaluation, we construct the held-out test dataset with 10% of the demonstrations. To improve the visibility of rendered frames, we adjust the camera view of the camera using the publicly available source codes.[6] We provide the dataset we used in our experiments[7].

**RoboNet.**    RoboNet (Dasari et al., 2019) is a large-scale real-world robotics dataset consisting of more than 160,000 high-resolution videos. Since there is no test set for RoboNet dataset, we follow the setup in Wu et al. (2021a) for constructing a held-out test dataset[8] of size 256. Following the setup in Wu et al. (2021a); Babaeizadeh et al. (2021), we train a video prediction model to predict ten future frames conditioned on two initial frames and ten future actions. For preprocessing the frames, we resize the original frames to $256 \times 256$ resolution frames without cropping. For downloading the dataset, we utilize the publicly available script.[9]

**Kinetics-600.**    Kinetics-600 dataset is a large-scale video dataset consisting of more than 400,000 videos of total 600 action classes. Following the setup in Clark et al. (2019); Luc et al. (2020), we train a video prediction model to predict future 11 frames conditioned on the first five frames. For downloading the dataset, we use the publicly available repository[10].

**BAIR Robot Pushing.**    BAIR Robot Pushing dataset (Ebert et al., 2017) consists of 43,264 training and 256 test videos. While BAIR dataset contains the information of actions robots take, common setup for evaluation on BAIR dataset is action-free (Clark et al., 2019; Weissenborn et al., 2020; Luc et al., 2020; Yan et al., 2021), where a video prediction model is trained to predict future 15 frames conditioned on the initial frame. For downloading and preprocessing the dataset, we utilize the publicly available script.[11]

**KITTI driving dataset.**    KITTI driving dataset (Geiger et al., 2013) is the dataset that contains a large number of high-resolution driving videos. However, for video prediction, we follow the setup in Lotter et al. (2017) and utilize 57 training videos and 3 test videos for evaluation. To avoid utilizing too similar video clips from the test dataset for evaluation, we follow the setup in Villegas et al. (2019) where 30-frame clip is extracted with the interval of 5 frames, which constructs a test of size 148. For comparison with baselines, following the setup in Villegas et al. (2019), we train a model to predict future ten frames conditioned on five frames and evaluate the model to predict future 25 frames conditioned on five frames. For downloading and preprocessing the dataset, we utilize the publicly available script.[12]

---

[5]https://github.com/rlworkgroup/metaworld/tree/master/metaworld/policies

[6]we use the of 9e3863d in https://github.com/rlworkgroup/metaworld.

[7]https://shorturl.at/acnxM

[8]We use the list of videos available at https://github.com/google-research/fitvid/blob/master/robonet_testset_filenames.txt

[9]https://gist.github.com/soskek/d762751ce0aef4b2c7cf0a1537917016

[10]https://github.com/cvdfoundation/kinetics-dataset

[11]https://github.com/wilson1yan/VideoGPT

[12]https://github.com/coxlab/prednet

## A.2 IMPLEMENTATION DETAILS OF HARP

**VQ-GAN.** The training of HARP consists of two-stages. First, for training a VQ-GAN model (Esser et al., 2021) we use the publicly available source code from the authors[13]. Following the suggestion of the authors, we trained a VQ-GAN model without a discriminator loss until it converges, then resume the training with a discriminator loss for total {300000, 500000, 150000, 30000} training steps with batch size of {24, 24, 320, 96} on Meta-World, RoboNet, BAIR Robot Pushing, and KITTI driving dataset, respectively. For Kinetics-600 dataset, we leverage the publicly available VQ-GAN model[14] pre-trained on ImageNet without training a VQ-GAN model from scratch. The size $k$ of codebook $C$ in (2) is {8192, 8192, 1024, 1024, 1024} for Meta-World, RoboNet, Kinetics-600, BAIR Robot Pushing, and KITTI dataset, respectively.

**Causal transformer.** Then we train a 12-layer Sparse Transformers (Child et al., 2019) to predict the discrete codes from VQ-GAN models in an autoregressive manner, by building on the publicly available source code[15] of VideoGPT (Yan et al., 2021). For conditioning on initial frames, we utilize the same architecture as in Yan et al. (2021) that utilizes ResNet architecture to extract the downsampled feature map. We utilize ResNet-18 architecture for all experiments. We train the model until it converges on Meta-World, BAIR, and KITTI datasets, but we cannot find the sign of overfitting on large-scale datasets of RoboNet and Kinetics-600. Specifically, we train the model for {50000, 80000, 100000, 85000, 30000} training steps on Meta-World, RoboNet, Kinetics-600, BAIR Robot Pushing, and KITTI dataset, respectively. We use the data augmentation with the magnitude of $m = 4$ on KITTI driving dataset, which has shown to be very effective for improving the performance (see Appendix B) for a learning curve with and without augmentation).

**Inference with top-$k$ sampling.** We utilize top-$k$ sampling (Fan et al., 2018) to improve the video prediction quality. For all datasets, we utilize $k = 10$. One major limitation of autoregressive prediction model is slow inference time. We utilize the implementation of Yan et al. (2021) that caches the previous key, values and utilize them for fast inference. In order to enable our model to extrapolate to unseen length of frames at evaluation time on KITTI dataset (*e.g.,* the model has to predict 25 frames instead of 10 frames the model is trained to predict), we first predict $T$ frames $\mathbf{x}_{c:T-1}$ conditioned on $c$ initial frames $\mathbf{x}_{1:c}$, then predict the next frames by (a) keeping $\mathbf{x}_{1:c}$ as conditioning frames and (b) giving the predicted last $T-1$ frames as inputs to the causal transformer model. We repeat this process until predicting all 25 future frames.

## A.3 IMPLEMENTATION DETAILS OF ACTION-CONDITIONED HARP.

In order to predict future frames conditioned on future actions on RoboNet dataset (Dasari et al., 2019), we condition the prediction on actions by adding the action embeddings to the embeddings of discrete codes. Specifically, we introduce a linear layer that processes raw actions to action embeddings with the same dimension of token embeddings, then add the action embeddings of time step $t + 1$ to token embeddings used for predicting the tokens of time step $t + 1$. At inference time, the inference procedure is exactly same as HARP except that future action embedding is added to the token embedding. We find that this simple modification to the original architecture enables HARP to predict future frames conditioned on actions. We provide the illustration of action-conditioned HARP in Figure 7.

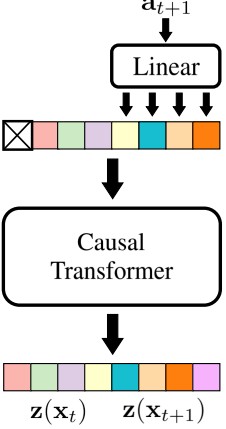

Figure 7: Action-conditioned HARP.

---

[13]https://github.com/CompVis/taming-transformers

[14]https://heibox.uni-heidelberg.de/d/8088892a516d4e3baf92/

[15]https://github.com/wilson1yan/VideoGPT

### A.4   IMPLEMENTATION DETAILS OF HARP FINE-TUNING.

In order to fine-tune the HARP model for solving multi-task imitation learning tasks on Meta-World (Yu et al., 2020), we first pre-train a model to predict future 11 frames conditioned on the first frame using the dataset from all 50 tasks (see Appendix A.1 for the details on the dataset). Then we fine-tune the model to predict expert actions by introducing a two-layer policy network on top of the causal transformer model. Specifically, we train a behavioral cloning policy to minimize the mean squared error between the predicted actions from a layer and ground-truth expert actions. In order to further improve the performance of all methods, we follow the idea of Dasari & Gupta (2020) to learn a inverse dynamics predictor that predicts the action given two consecutive frames. We train all methods for 20,000 steps with the batch size of 20 and data augmentation of magnitude $m = 4$. For CNN-based methods, we use ResNet-50 for feature extractor, and use 4-layer LSTM[16] for CNN-LSTM, and 12-layer causal transformer for CNN-Transformer.

## B   EFFECTS OF AUGMENTATION

Figure 8 shows the test error during the training of HARP on KITTI driving dataset (Geiger et al., 2013). One can see that the test error from a model trained without augmentation increases after initial $\sim 2500$ training steps, which is a sign of overfitting. However, we observe that the test error from a model trained with data augmentation keeps decreasing throughout training until it converges, which shows the effectiveness of data augmentation for learning video prediction models, similar to the observation in Babaeizadeh et al. (2021). One notable detail here is that HARP overfits to KITTI training dataset with much fewer number of parameters (*i.e.,* 89M) when compared to FitVid that utilizes 303M parameters.

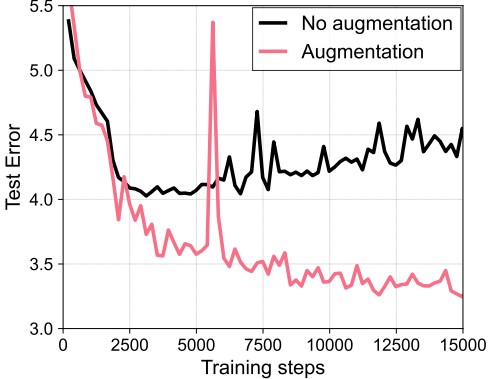

Figure 8: Test loss during the training of HARP with and without data augmentation on KITTI driving dataset (Geiger et al., 2013). We observe that the model easily overfits to the training dataset, and data augmentation helps mitigate the overfitting problem.

---

[16]We try more deeper networks to match the number of trainable parameters, but we find that deeper LSTM networks are very unstable to train. Hence we search over the layers of {2, 4, 8, 16} and report the best results achieved with 4-layer LSTM.

## C   VIDEO PREDICTIONS ON BAIR AND KITTI

We provide the future frames predicted by HARP on BAIR Robot Pushing (Ebert et al., 2017) and KITTI driving dataset (Geiger et al., 2013). The model is trained to predict future 15 frames conditioned on the first frame on BAIR dataset. In the case of KITTI dataset, the model is trained to predict future 10 frames conditioned on the five frames, and evaluated to predict future 25 frames conditioned on the five frames.

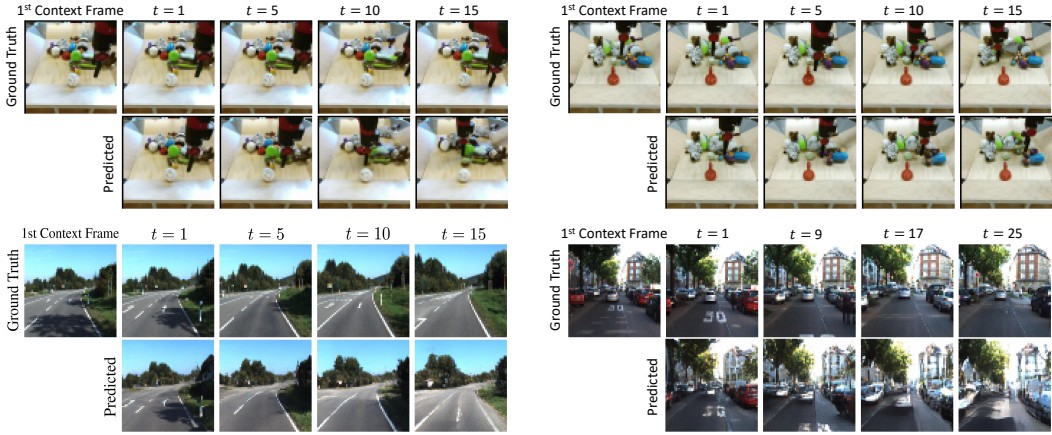

Figure 9: $64 \times 64$ future frames predicted by HARP on BAIR dataset (up) and KITTI dataset (bottom). We observe that our model can predict diverse future frames with stochastic sampling on BAIR dataset, and can deal with the partial observability of the KITTI dataset.

## D   MORE VIDEO PREDICTIONS ON META-WORLD

We provide more videos predicted by HARP on Meta-World dataset (Yu et al., 2020). The model is trained to predict future 11 frames conditioned on the first frame.

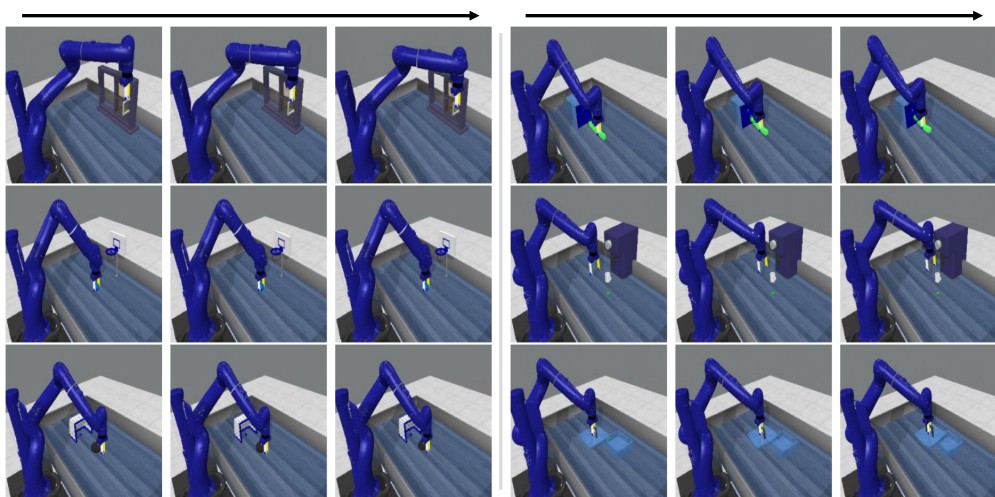

Figure 10: $64 \times 64$ future frames predicted by HARP on Meta-World dataset.

# E    MORE VIDEO PREDICTIONS ON ROBONET

We provide more videos predicted by HARP on RoboNet dataset (Dasari et al., 2019). The model is trained to predict future ten frames conditioned on the initial two frames and future ten actions.

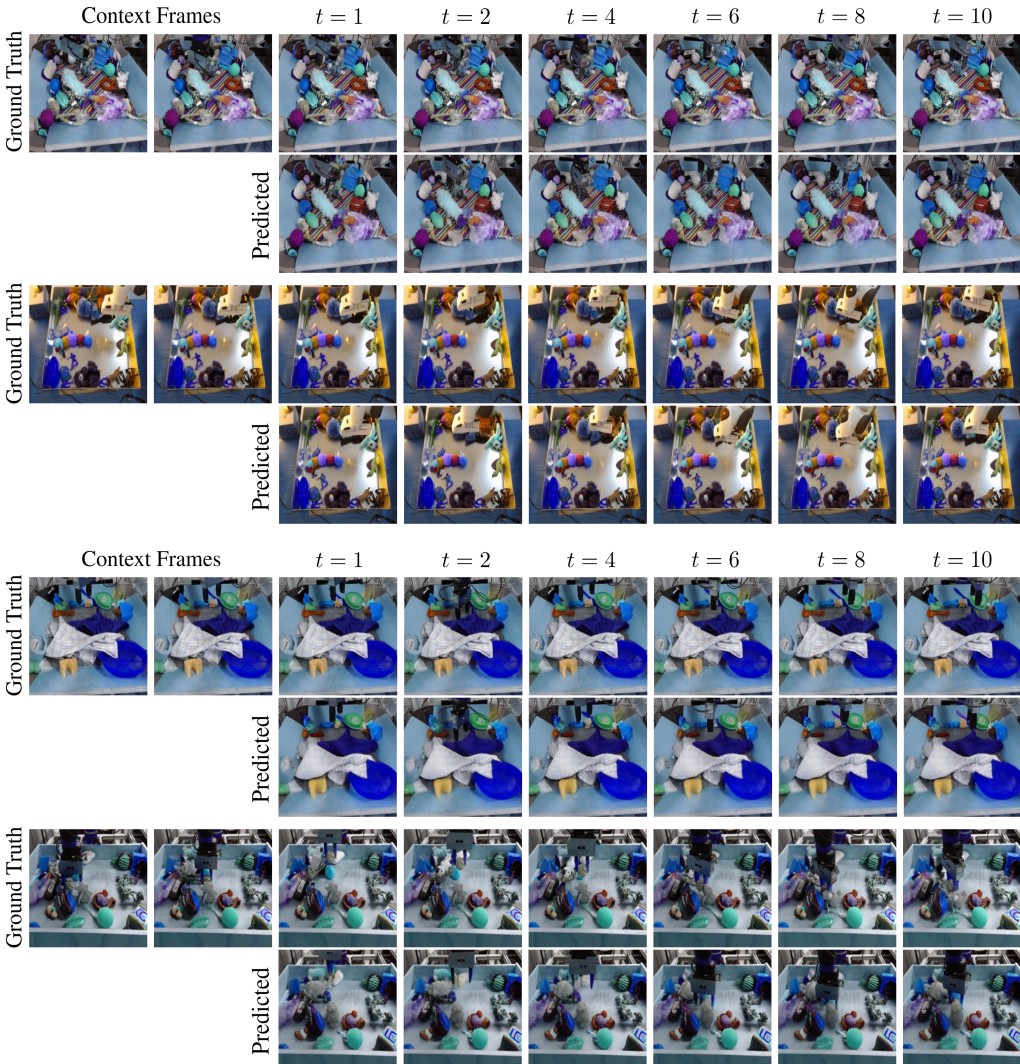

Figure 11: $256 \times 256$ future frames predicted by HARP on RoboNet dataset.

## F   MORE VIDEO PREDICTIONS ON KINETICS-600

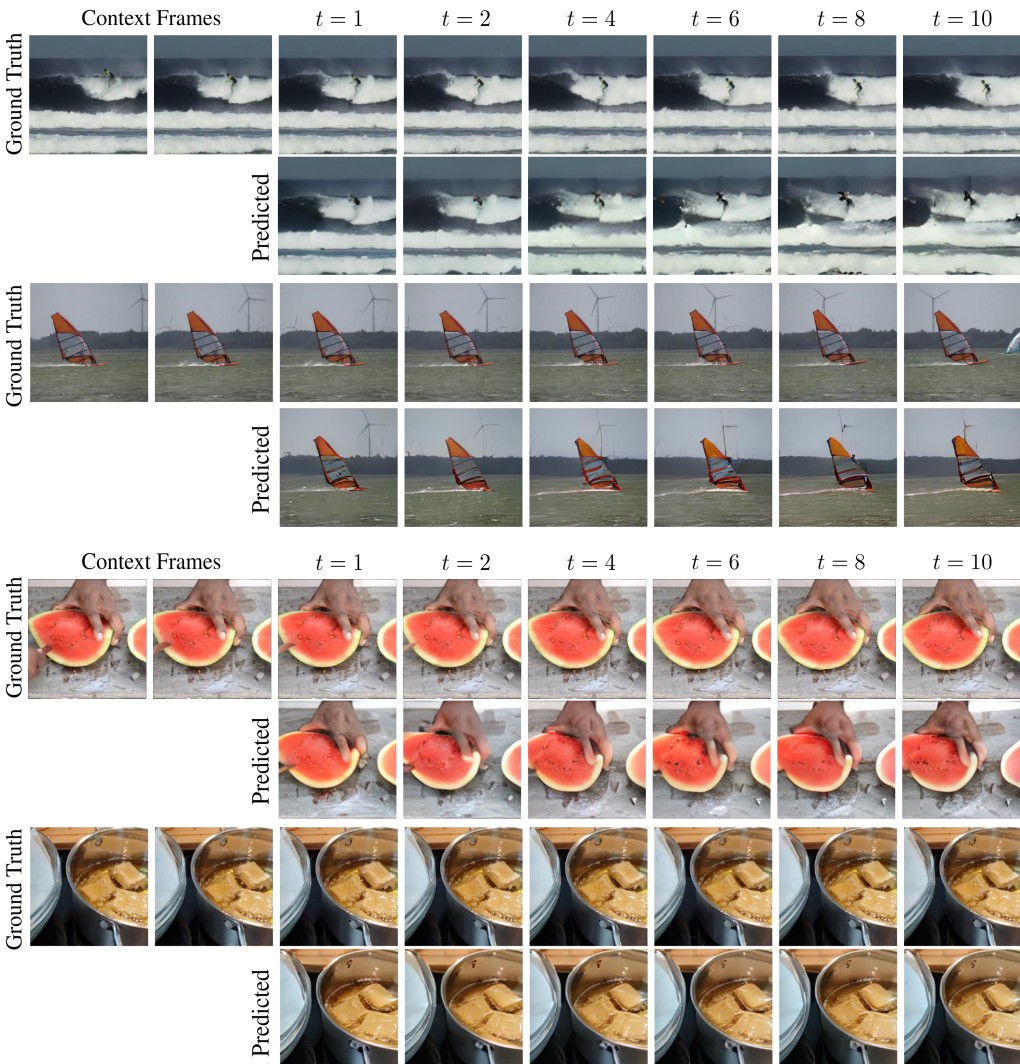

Figure 12: $256 \times 256$ future frames predicted by HARP on Kinetics dataset.

# G    ATTRIBUTION

Figure 5 (top):
"Windsurfing Session 2013_Iballa Moreno" by morenotwins. Accessible here

Figure 5 (bottom):
"How to fry an egg in 5 simple steps" by What's Gaby Cooking. Accessible here

Figure 6 (right):
"Prepare fruit cutting to pets #Shorts 4" by AP STUDIO. Accessible here

Figure 12 (first):
"Windsurfing" by Dmitry Rudnev. Accessible here

Figure 12 (second):
"Eric Cornelissen Windsurfing September 11, 2017" by Ron Van Dijk. Accessible here

Figure 12 (third):
"Smart way to cut Fruits #6#" by MR. BEING SMART. Accessible here

Figure 12 (fourth):
"How to make the Perfect Crunchy Half Cakes ( Kangumu ) ‖‖ Jinsi ya kupika Half Cakes za kupasuka." by Ayleen's Food & Vlogs. Accessible here

