# OpenReview forum: "Autoregressive Latent Video Prediction with High-Fidelity Image Generator"
_ICLR.cc/2022/Conference — ICLR 2022 Submitted_

### Official Review · Reviewer_pKCo · 2021-10-22

**Correctness:** 2
**Technical Novelty And Significance:** 2
**Empirical Novelty And Significance:** 2
**Recommendation:** 3
**Confidence:** 4

**Main Review:**

**Strengths**

* The paper is well written and easy to read and the proposed framework is conceptually simple.
* Authors show qualitative results on 256x256 images which is quite challenging.
* The reported results on KITTI are strong.

**Weaknesses**

* The main technical novelty seems to be in the usage of VQ-GAN model (instead of the VQ-VAE model) to produce latent codes. But there is no ablation that quantifies the impact of this design choice. The authors should replace VQ-GAN with the standard VQ-VAE model and report the improvements (if any).
* The paper claims that the prior related works are limited to generation of 128x128 images. But it is unclear what design decisions allow HARP to generate 256x256 resolution images. In terms of complexity, the proposed HARP should be similar to LVT (Rakhimov et al 2021) and VideoGPT (Yan el atl 2021) since both rely on attention-based architectures in latent space. Is it that the prior works could generate 256x256 images but with poor fidelity due to the reliance on VQ-VAE (instead of VQ-GAN)? This is simple to verify by just replacing the VQ-GAN with VQ-VAE and reporting the results on 256x256 images.
* The authors should report their quantitative results on Kinetics600 64x64 to provide another point of comparison. Additionally, they should report their numbers on Kinetics600 256x256 with and without ImageNet pre training as a future baseline.
Further, [Clark et al 2019] report results on high resolution Kinestics600 in Table 1 of their paper, which should be compared to.
* The final comparison in Table 2 is reported using top-k sampling, which is unfair to both LVT and VideoTransformer as top-k sampling is readily applicable to these models as well. After removing top-k sampling, the performance of HARP is comparable to VideoGPT.
* What is “number of layers” in the Results Section? Is it the number of layers in VQ-GAN or the 1-D Transformer?
* How was HARP tuned? The datasets in Section 5.2 only talk about the train and test split but no validation split?
* Can the authors provide some ablations on the codebook size? Since it is specifically stated that “given that the number of discrete encodings required for future prediction is very large, e.g., 2,560 on RoboNet (Dasari et al., 2019) up to 6,400 on KITTI dataset (Geiger et al., 2013) in our experimental setup.”. How were these codebook numbers arrived at?


**Summary Of The Paper:**

The paper proposes an autoregressive latent video prediction model with two components:
* An image generation model (VG-GAN) is trained to compress an image into 2-D latent codes.
* A 1-D autoregressive transformer generates the next latent code given the flattened history of the previous latent codes, i.e (H*W*T). This latent code is then transformed into the next frame of the sequence by the decoder of the VQ-GAN model.
* Quantitative Results are shown on 64x64 Images on the BAIR Robot Pushing dataset and KITTI driving dataset.
* Some performance improvement is shown using top-k sampling on the BAIR dataset and KITTI dataset and data-augmentation (pixel-translation) on the KITTI dataset.
* Qualitative Results are shown on high-resolution images (256x256) on the RoboNet, MetaWorld and Kinetics600 datasets.


**Summary Of The Review:**

My initial rating is reject. The technical novelty of the paper is arguably only the VQ-GAN encoder. The authors should show more empirical evidence that this component is crucial for their results on video prediction. This can be done via ablations and comparisons with prior work (as described in my weaknesses section)

---

### Official Review · Reviewer_6x6m · 2021-10-29

**Correctness:** 4
**Technical Novelty And Significance:** 2
**Empirical Novelty And Significance:** 2
**Recommendation:** 3
**Confidence:** 4

**Main Review:**

The proposed method is an interesting contribution for the community, since it reduces the complexity of learning video prediction models with appealing results. The paper contains a simple, actionable and effective methodology which is supported by the experiments. High-resolution ($256 \times 256$) forecasting is one of the main advantage of HARP as it is rare in the literature, especially using reasonable computing power. The empirical evaluation partly illustrates the benefits of the method, thanks to results matching the state of the art with limited parameter count. Overall, the paper is well written and easy to read, and experiments seem to be reproducible.

However, I find the contribution of this paper to be insufficient for two main reasons.

### Novelty

The first limitation deals with the novelty of the method. As acknowledged by the authors, the proposed two-stage training has already been proposed in prior works: Rakhimov et al. (2021) and Yan et al. (2021). While these articles are recent, the reviewed paper is not concurrent and clearly builds on them. Indeed, to my understanding, the proposed method amounts to instantiating e.g. VideoGPT (Yan et al., 2021) with another more powerful image generator of the same nature (VQGAN instead of VQVAE) and the same transformer architecture. This significantly reduces the originality of the approach.

### Experiments

The second limitation lies in the experimental evaluation which does not compensate the lack of novelty, thereby limiting the significance of the overall contribution. The issues are multiple and described below.

Regarding the forecasting experiments on Meta-World, RoboNet and Kinetics-600, presented results seem satisfying but there lacks some point of comparison in the paper to assess their significance. This could consist in performance comparison with another baseline (like Video transformer which was tested on Kinetics in the original article or another autoregressive model like VideoGPT) or, because of scalability or computational power issues, a more detailed analysis of the computational advantages of HARP against the other baselines. The only provided comparison is on parameter count vs. a single other model, which is insufficient (cf. also the forthcoming remarks on parameter count). I do believe that HARP is computationally appealing, but a factual comparison is needed to confirm this impression.

Regarding the forecasting experiments on BAIR and KITTI, the presented results do not allow us to unambiguously conclude about the advantages of HARP. The only comparison with a similar model (VideoGPT) is not conclusive, since the parameter count and performance of both models are similar. More generally, there are fairness issues in the comparisons that prevent the reader to correctly assess the performance of HARP. Indeed, top-$k$ sampling and data augmentation are beneficial to the model but it is unclear how the other models could benefit from them as well and how this would change the performed comparison. For example, the authors may discuss the impact of data augmentation on the considered baselines, as well as the links of top-$k$ sampling with temperature reduction in generative models like in VideoFlow (Kumar et al., 2021) which could similarly improve the performance of the other models.

Finally, I fear that the sole indication of parameter count to evaluate the efficiency of a model may be insufficient for these comparisons, as it is an imperfect proxy of the actual capacity of computational requirements of a model (unless the numbers of parameters are orders of magnitude different). More information like GPU, memory and training time requirements for the baselines would circumvent this issue. Ideally, a study on the influence of the number of parameters on the performance of HARP and selected baselines would greatly improve the comparison.

Less importantly, I would optionally suggest the authors to complement the numerical evaluation with the standard PSNR metric, like e.g. Denton et al. (2018) and Lee et al. (2018). Despite its flaws, PSNR is complementary to the perceptual scores FVD and LPIPS as it can better reveal dynamics prediction errors. Considering a synthetic dataset like Moving MNIST, that remains challenging for long-term prediction, would also help in this regard to confirm the ability of transformers to accurately predict dynamics on pretrained representations as proposed in the paper.

**Summary Of The Paper:**

This paper introduces a new video prediction model, HARP, whose training operates in two stages: pretraining an image autoencoder on the frames with discrete latent states, then learning a transformer predictor on the resulting latent space instead of the pixel space. The authors complement this procedure with data augmentation during training and a modification of latent code sampling during inference in order to improve the performance of the model. This experimentally results in an efficient training procedure, with a model that is demonstrated to be able to generate high-resolution videos and match or outperform the state of the art in the domain on multiple datasets and tasks.

**Summary Of The Review:**

Considering the two important limitations described above, I find this paper to be below the acceptance threshold. Given the limited technical novelty, the empirical evaluation should be stronger for this paper to be accepted. I believe that such an improvement could make it ready for publication.

I understand that the suggested improvements are significant and it may be difficult for the authors to properly address them during the rebuttal. Nonetheless, I would be pleased to increase my score, should my concerns or misunderstandings be clarified during the discussion.

### Post-Rebuttal Update

I acknowledge the authors' response and would like to thank them for their answers. As stated by the authors themselves, the necessary improvements could not have been implemented during the rebuttal, which is understandable. Therefore, I choose to maintain my score.

---

### Official Review · Reviewer_3gJ8 · 2021-11-02

**Correctness:** 4
**Technical Novelty And Significance:** 1
**Empirical Novelty And Significance:** 2
**Recommendation:** 5
**Confidence:** 4

**Main Review:**

**Strengths:** I think this work fits well into the recent literature on generative models, which divides the learning process of perceptually highly redundant data (such as images, audio, or videos) into a two-stage procedure (e.g., [1, 2, 3]), where the first stage is trained to achieve maximum compression while still yielding perceptually close representations, and the actual generative model is then trained on this compressed representation in a second stage. This has the distinct advantage that the generative model does not have to "relearn" the compression itself (which is particularly difficult in mode-covering approaches such as AR likelihood models) and can focus on salient high-level features. However, this also reveals a weakness of the present work: the "compressibility" is only exploited for single frames, but not for the temporal dimension, which is particularly wasteful for video data.

**Weaknesses:** The novelty of _HARP_ over previous works is quite limited: There is a large body of work such as [4,5,6] that uses a two-stage procedure for video generation.  The main advantage over VideoGPT is that HARP does not rely on a hierarchical representation and uses a VQGAN instead of a VQVAE. Also, I don't quite understand why top-k sampling is "sold" so much; it is an established technique in dealing with autoregressive models. What about other pruning techniques like nucleus sampling or temperature rescaling of logits? In addition, the work should also qualitatively demonstrate sensitivity to the specific value of $k$. Finally, a quantitative analysis of the "high resolution" experiments ($256 \times 256$ px) is missing.

**Mixed Comments:**
- The technical details regarding the baseline models in Sec. 5.3 are very sparse. What does "on top of the transformer model" mean? How exactly does the CNN+Transformer model differ from the VQ+Transformer model? Is it possible to train only the policy network while fixing the (pre-trained) transformer model?
- Does the VQGAN's codebook collapse affect training, and if so, how is it taken into account?
- How long does it take to sample the different models? How does it compare to other approaches?
- What is $m$ in Sec. 5.2?
- Sec. 5.2 and Tab.2 heavily rely on the fact that HARP uses less parameters than some other models, but does not explicitly state why this is useful. See also [7] on this "efficiency misnomer".

__References__
- [1]: Esser, Patrick, Robin Rombach, and Bjorn Ommer. "Taming transformers for high-resolution image synthesis." _Proceedings of the IEEE/CVF Conference on Computer Vision and Pattern Recognition_. 2021.
- [2]: Iashin, Vladimir, and Esa Rahtu. "Taming Visually Guided Sound Generation." _arXiv preprint arXiv:2110.08791_ (2021).
- [3]: Mentzer, Fabian, et al. "High-fidelity generative image compression." _arXiv preprint arXiv:2006.09965_ (2020).
- [4]: Yan, Wilson, et al. "VideoGPT: Video Generation using VQ-VAE and Transformers." _arXiv preprint arXiv:2104.10157_ (2021).
- [5]: Rakhimov, Ruslan, et al. "Latent video transformer." _arXiv preprint arXiv:2006.10704_ (2020).
- [6]: Dorkenwald, Michael, et al. "Stochastic Image-to-Video Synthesis using cINNs." _Proceedings of the IEEE/CVF Conference on Computer Vision and Pattern Recognition_. 2021.
- [7]: Dehghani, Mostafa, et al. "The Efficiency Misnomer." _arXiv preprint arXiv:2110.12894_ (2021).


**Summary Of The Paper:**

This paper proposes a very simple method (dubbed _HARP_) for video synthesis: By relying on a pretrained VQGAN which encodes (and decodes) each frame seperately to (and from) a highly compressed representation, a (sparse) transformer model can be trained as an autoregressive generative model on the sequence of discrete image representations. The paper shows that despite its simplicity, the proposed approach can compete with the state of the art and produce high quality video at $256 \times 256$ resolution. As a side contribution, it is shown that the proposed model can be fine-tuned towards multi-task imitation learning.

**Summary Of The Review:**

The main advantage of the paper is its simplicity, and that is what distinguishes it from other approaches, i.e. it is simpler, but achieves better or close to SOTA results. The demonstration that such a simple approach can give (near) top results is definitely interesting.
However, the novelty value compared to previous approaches is very limited. All in all, I'm leaning towards a rejection because I'm not convinced that the results of the paper warrant a full conference publication and might be better suited for a workshop, but I'm curious to see what the other reviewers think.

---

### Official Review · Reviewer_JjT4 · 2021-11-03

**Correctness:** 4
**Technical Novelty And Significance:** 2
**Empirical Novelty And Significance:** 2
**Recommendation:** 3
**Confidence:** 4

**Main Review:**

Strengths:
   + The paper is well written and easy to follow.
   + The idea of leveraging pre-trained image generator for high-fidelity frame outputs while using transformer to focus on learning video dynamics is well motivated.
   + Experiments on a wide range of video prediction benchmarks demonstrate the proposed method can generate visually good results and performs well compared to state-of-the-art methods on this challenging problem.

Weaknesses:
   + While I appreciate the good results achieved by the proposed method, my major concern is that the novelty of it is somewhat limited. The idea of combining image generator with latent space auto-regressive model has been widely explored in previous works ([1], [2], [3], [4]). At the high level, the idea of the proposed method is similar to [1], only with VQ-VAE replaced by VQ-GAN. It is not surprising that the visual quality of the prediction results improves with a better image generator and I don't feel that it provides much novelty by itself.
   + I also feel that the paper lacks in-depth discussion on the relation between the proposed method and other relevant works. In particular, the overall framework followed by the proposed method, including the top-k sampling strategy, is very similar to that of [5]. Yet there is no discussion on [5] except mentioning of the use of the same VQGAN model. Can the proposed method be considered a direct extension of [5] from image generation to video generation? Is there any fundamental challenge in doing that extension? and is there any fundamental change that the author incorporated to address such challenge?

[1] Wilson Yan, Yunzhi Zhang, Pieter Abbeel, and Aravind Srinivas. Videogpt: Video generation using
vq-vae and transformers. arXiv preprint arXiv:2104.10157, 2021.

[2] Dirk Weissenborn, Oscar Tackstr ¨ om, and Jakob Uszkoreit. Scaling autoregressive video models. In ¨
International Conference on Learning Representations, 2020.

[3] Ruslan Rakhimov, Denis Volkhonskiy, Alexey Artemov, Denis Zorin, and Evgeny Burnaev. Latent
video transformer. In International Joint Conference on Computer Vision, Imaging and Computer
Graphics Theory and Applications, 2021.

[4] Yu Tian, Jian Ren, Menglei Chai, Kyle Olszewski, Xi Peng, Dimitris N Metaxas, and Sergey
Tulyakov. A good image generator is what you need for high-resolution video synthesis. In
International Conference on Learning Representations, 2021.

[5] Patrick Esser, Robin Rombach, and Bjorn Ommer. Taming transformers for high-resolution image
synthesis. In Proceedings of the IEEE/CVF Conference on Computer Vision and Pattern Recognition, 2021.

**Summary Of The Paper:**

This paper presents a video prediction method that combines a transformer-based auto-regressive model with a pre-trained image generator for high-resolution frame generation. The image generator is based on VQGAN. The auto-regressive model is trained to predict the discrete VQGAN's latent codes which is then used by the image generator to generate patches in the video. Experiments on different video prediction benchmarks demonstrate the effectiveness of the proposed method for high-resolution (256x256) video prediction.

**Summary Of The Review:**


In general, I found the proposed method effective in generating visually good video prediction results. However, I found the technical contribution of the paper somewhat limited. Each technical component is not new and the way of combining them has also been explored before, which limit the novelty value of the paper. Please see the previous box for more strengths and weaknesses details.

---

### Official Review · Reviewer_3FiN · 2021-11-03

**Correctness:** 3
**Technical Novelty And Significance:** 3
**Empirical Novelty And Significance:** 3
**Recommendation:** 5
**Confidence:** 4

**Main Review:**

Strengths:
* The paper is well-written and easy to follow.
* The idea is simple and clear.
* It is interesting to see an ImageNet pre-trained VQ-GAN can be used for predicting Kinetics-600 videos

Weaknesses:
* Loss functions are missing.
* Though using fewer parameters, the proposed method, HARP, doesn’t outperform baselines on the BAIR dataset. As in table 2(b), simply increasing the transformer layers can improve the performance, it is interesting to see if HARP can outperform the baselines by increasing the parameters to the same level (~300M).
* The quantitative evaluation of video prediction is only conducted on BAIR and KITTI datasets, as KITTI is a small-scale dataset, it’s better to evaluate HARP with more datasets.
* There are domain gaps between ImageNet images and Kinetics-600 videos, how does HARP deal with such domain gaps? Is finetuning needed? Also in the paper, the predicted videos are only shown with some selected categories. How about the performance on the whole Kinetics-600 datasets? (e.g. examples on more categories and/or FVD scores on the whole dataset)

**Summary Of The Paper:**

This paper introduces HARP for high-resolution video prediction. Specifically, they leverage a pre-trained image generator (VQ-GAN) and fix it in the video prediction stage. The predicted codes are generated by a progressive causal transformer. Other techniques, like top-k sampling and data augmentation, are further employed to improve the diversity and quality. Experiments show it outperforms baselines on several datasets.

**Summary Of The Review:**

This paper introduces a new method for high-resolution video prediction, however, the experiments are not sufficient, I believe more experiments are needed to justify the effectiveness of the proposed method.

---

### Author Response · Authors · 2021-11-23
**Response to Reviewers**

We sincerely thank all the reviewers for their helpful and constructive comments on our paper. We found that the reviewers raised the concerns on the limited technical novelty and empirical evaluation does not compensate the weakness. Unfortunately, we found that it is difficult to run all the experiments to address the concerns of the reviewers on experimental evaluation during the rebuttal period. However, we would like to answer the questions, respond to the suggestions you provided, and clarify the details for better understanding of our manuscript.

---

**Q1:** Missing loss function

**A1:** Our loss function is designed to maximize equation (6), and specifically it is implemented to minimize the cross entropy loss using the discrete codes extracted from the VQ-GAN as labels.

---

**Q2:** Domain gap between ImageNet and Kinetics-600

**A2:** Thanks for the interesting question. We found that reconstructions from ImageNet-pretrained models do not perfectly work on Kinetics-600 frames, e.g., the model sometimes tries to generate animals based on the color and shape of the input frames as the model is trained on ImageNet dataset consisting of a lot of animal images. We expect finetuning can be helpful for generating more realistic future frames on Kinetics-600 dataset.

---

**Q3:** VQGAN’s codebook collapse

**A3:** We empirically found that there was no codebook collapse during the training of VQGANs on our considered datasets.

---

**Q4:** Sampling time

**A4:** Even though we reduced the sampling time using the caching scheme used in [Yan et al., 2021], it still takes much more time for generating videos compared to the other video prediction methods, which is one of limitations of our method. It would be interesting direction to develop a method that can generate high-fidelity video efficiently at inference time.

---

**Q5:** What is $m$?

**A5:**$m$ is the magnitude of the data augmentation we used, e.g., how much the pixels in the frames are shifted.

---

**Q6:** Usefulness of the number of parameters for measuring the efficiency

**A6:** Thanks for the pointer to [Dehghani et al., 2021]. We will consider and incorporate other metrics to evaluate the efficiency of our method, including the computing resources used for each method.

---

**Q7:** Why is top-$k$ sampling is so emphasized?

**A7:** While we agree on that top-$k$ sampling is an widely used technique in autoregressive models, our work aims to show the importance of such sampling schemes in the context of video prediction, given that length of tokens to predict is often significantly long compared to the commonly considered sequence length in other domains.

---

**Q8:** Other metrics like PSNR

**A8:** Thanks for helpful suggestion. We will try to include other metrics like PSNR.

---

**Q9:** Clarification on “number of layers”

**A9:** By “number of layers”, we mean the number of layers for Transformer layers.

---

**Q10:** How was HARP tuned?

**A10:** Thanks for pointing this out. We used the model whose training loss is the lowest, because we found no sign of overfitting on BAIR, and KITTI when we use data augmentation. We expect that performance can be further improved with more training, but couldn’t run the experiments until the loss converges due to the limited resources.

---

**Q11:** Clarification on the number of discrete encodings

**A11:** The number of discrete encodings required for future prediction is not related to the number of codes in a VQ-GAN codebook. For example, 2,560 on RoboNet and 6,400 on KITTI datasets are calculated as [height of downsampled feature map * width of downsampled feature map * number of future frames to predict], i.e., 2,560 = [16 * 16 * 10] and 6,400 = [16 * 16 * 25], so their numbers are not related to the codebook sizes.

---

[Yan et al., 2021] Wilson Yan, Yunzhi Zhang, Pieter Abbeel, and Aravind Srinivas. Videogpt: Video generation using vq-vae and transformers. arXiv preprint arXiv:2104.10157, 2021.

[Dehghani et al., 2021]: Dehghani, Mostafa, et al. "The Efficiency Misnomer." arXiv preprint arXiv:2110.12894 (2021).

---

Thanks again for all the reviewers' time to review our manuscript.

Authors

---

### Decision · Program_Chairs · 2022-01-20

**Decision:**

Reject

**Comment:**

The submission proposes to learn a causal transformer model over a pretrained VQ-GAN representation to generate videos. While the paper is well written and clear, proposing a simple idea, the novelty of this method is not well explained compared to pre-existing publications see ([reviews JjT4](https://openreview.net/forum?id=K-hiHQXEQog&noteId=22EyPvpodh), [3gJ8](https://openreview.net/forum?id=K-hiHQXEQog&noteId=9ikn_nBC_Sf), [6x6m](https://openreview.net/forum?id=K-hiHQXEQog&noteId=zCX9VP8I5uL), and [pKCo](https://openreview.net/forum?id=K-hiHQXEQog&noteId=uRgHWX4C5yX)) especially since it's lacking [ablation](https://openreview.net/forum?id=K-hiHQXEQog&noteId=uRgHWX4C5yX) or comparative (see [reviews 6x6m](https://openreview.net/forum?id=K-hiHQXEQog&noteId=zCX9VP8I5uL), and[pKCo](https://openreview.net/forum?id=K-hiHQXEQog&noteId=uRgHWX4C5yX)) experiments.
The authors have expressed their consideration of reviewers requests but will not satisfy them in time for this conference. Therefore I am currently recommending this submission for rejection.